# The prevalence of pain catastrophising in nulliparous women in Nepal; the importance for childbirth

Carol J. Clark[ID][1]*, Sujan Babu Marahatta[2,3,4,5], Vanora A. Hundley[ID][1]

1 Faculty of Health and Social Sciences, Centre of Midwifery and Womens Health, Bournemouth University, Bournemouth, United Kingdom, 2 Manmohan Memorial Institute of Health Sciences Kathmandu, Kathmandu, Nepal, 3 Bournemouth University, Bournemouth, United Kingdom, 4 Nepal Open University, Lalitpur, Nepal, 5 Department of Public Health Sciences, University of California, Davis, California, United States of America

* cclark@bournemouth.ac.uk

**Data Availability Statement:** The data underlying the results presented in the study are available from Bournemouth University BORDaR https://doi.org/10.18746/bmth.data.00000365

## Abstract

In Lower-Middle-Income-Countries women are encouraged to present at a birthing facility for skilled care, but attending early can be associated with additional harm. Women admitted in latent labour are more likely to receive a cascade of unnecessary interventions compared with those attending a birthing facility during active labour. One reason that women present early is pain, with higher rates of admission among those who pain catastrophise. The aim of this study was to explore the prevalence of pain catastrophising in nulliparous women in Nepal and to identify predictors for pain catastrophising. A cross sectional study was conducted using a semi-structured survey. The survey was completed by 170 women (18–32 years) in one higher education institution in Kathmandu. The survey included the pain catastrophising scale (PCS), current and previous pain and information about period pain, socio-demographic variables of age, ethnicity, and religion. The prevalence of pain catastrophising reported at a cut off score of PCS≥20 was 55.9% and at a cut off score of PCS≥30 was 17.1%. All women with a PCS ≥30 reported having painful periods. Those with a PCS≥20 were four times [95%CI 1.93–8.42] more likely to report painful periods affecting their daily activities (p<0.001) and those with PCS≥30 three times [95%CI1.10–10.53] more likely (p<0.05). In both cases ethnicity and age were not associated. Women with higher PCS were less likely to take pain medication. A high prevalence of pain catastrophising was reported. It is important to understand how women's previous negative experiences of pain and pain catastrophising are perceived and if they are contributing to the rise in obstetric intervention, particularly caesarean births, in Nepal. We recommend repeating this study with a larger sample representing a more diverse population.

## Introduction

Facility births in low-and-middle-income countries (LMICs) has been encouraged over the last two decades [1] with the aim of reducing maternal and new-born morbidity and mortality

**Funding:** The funders had no role in the study design, data collection and analysis, decision to publish, preparation of manuscript The study had no grant number as it was supported through internal quality related funding. Initials of author CJC Bournemouth University. https://www.bournemouth.ac.uk/

**Competing interests:** The authors have declared that no competing interests exist.

[2]. Nepal has made significant progress in increasing the proportion of women accessing facilities for birth. Although this trend was reversed during the lockdown period (March-May 2020) resulting in a sharp increase in maternal mortality (24 deaths in the two-month lockdown period compared with 80 for the previous year) [3].

It could be argued that there is a positive correlation between facility births and maternal and neonatal outcomes; however, varying quality of care within institutions does not always lead to improved outcomes [4] especially if facilities are unable to deal with obstetric emergencies. This is a particular concern is relation to early labour. Women who are admitted into hospital in the latent phase of labour (the time before labour is established) are more likely to experience unnecessary interventions compared with those admitted in the active phase of labour [5–7]. Although most of this research has been conducted in high resources settings, there is evidence that this also occurs in LMICs [8]. In these settings unnecessary intervention not only depletes scarce resources but also puts the woman at risk of increased morbidity. There is evidence in LMIC settings that women who perceive themselves to be at 'high-risk' are more likely to present to a birthing facility [4]. This perception relating to 'high risk' might be because of fear of childbirth, emotional distress, and their perceived ability to manage their pain. For example, if women know they will be offered the options of pain relief they may be more likely to seek to give birth in a facility as this may reduce their fear around childbirth [9].

It is well acknowledged that women seek admission to hospital in the latent phase of labour because they are fearful of childbirth, anxious and because they have pain [10]. Perceptions of significant pain are not new to women as many women experience discomforting pain regularly as part of their menses (period pains). This pain can be significant enough to reduce physical and social activities as well as quality of life [11]. Painful periods are recognised by women as 'normal' and over time they build strategies to manage their pain with the knowledge that the pain will eventually subside. Painful periods have been found to be associated with an amplified pain perception response [12] and may be one factor which contributes to their pain response during labour.

Women in high resource settings report negative experiences of managing their pain at home during the latent phase of labour as they feel neglected, unsupported, and anxious [13]. As their anxiety builds it is likely women feel more pain. For nulliparous (a woman who has not given birth to a child) women their anxiety is further contributed to as they are not confident about when 'the right time' will be, what the birthing process will involve and when the pain will subside [10].

What matters most to women is to be able to experience a normal birth with good outcomes for themselves and their baby [1]. In Nepal women's abilities to manage their labour and feel in control were negatively associated with their perceptions of labour pain [14]. Pain perception involves neuropsychological processes beyond the painful sensation, these include a cognitive appraisal of the meaning of the pain which is influenced by emotion and psychophysiological and behavioural reactions [15]. Interpretation of pain can be considered on a continuum where pain perception may be considered as manageable through to completely unmanageable. For those who perceive the pain to be manageable they may be able to modulate their pain response through pharmacological treatments or non-pharmacological strategies. In contrast at the other end of the spectrum where the pain intensity is perceived as unmanageable a person may experience catastrophic thoughts about their pain. These uncontrollable negative thoughts and feelings of helplessness that some individuals experience when they feel pain is referred to as pain catastrophising. Pain catastrophising can be measured using the Pain Catastrophising Scale (PCS) [16] and more recently this has been translated and validated for use in Nepal [17]. Between country differences in pain beliefs, coping and

catastrophising but it is not clear if the findings are clinically significant, and the data presented was not gender specific [18].

Women in the latent phase of labour report pain catastrophising to be associated with a fear of being overwhelmed by pain, resulting in women being more likely to request pain relief [19]. It is also suggested that fear in childbirth and pain catastrophising may be driving preferences for operative births [20–22].

Pain catastrophising is not just a phenomenon of childbirth but might be considered an inherent trait associated with a history of pain experiences [23]. In a UK study of healthy students nearly half the participants reported pain catastrophising at a cut-off score of PCS ≥20 and over a fifth reported pain catastrophising at a PCS ≥30 [24]. Unsurprisingly the study showed that pain catastrophising in these young women was significantly associated with fear of pain and pain-related anxiety. In Nepal pain catastrophising has been recognised as a characteristic that influences post-operative pain intensity. in addition, it was reported that women were found to have significantly higher PCS scores than men and a higher PCS was correlated with increases in requirements for pain relieving medication [25]. The pattern of labour pain differs between nulliparous and multiparous women, and it is well documented that pain scores are higher in the nulliparous compared with multiparous woman especially if there has been no antenatal education [26]. There have been no studies exploring the prevalence of pain catastrophising in healthy young nulliparous women with a view to exploring how this might affect childbirth and their pain management requirements. This study was designed to address the gap in the literature. The aim of this study was to explore the prevalence of pain catastrophising in nulliparous women in Nepal and to identify predictors for pain catastrophising.

## Methods

### Study setting

The study was conducted with one higher education institution (HEI) in Kathmandu, the capital of Nepal. The institution provides education to a range of different disciplines including nursing, pharmacy, public health, and laboratory sciences.

### Study design and participants

A cross sectional study was conducted using a semi-structured survey. Data were collected in October and November 2022 using one validated questionnaire [16,17], additional questions relating to pain and demographic information with a convenience sample of nulliparous women of reproductive age.

In total, 170 women aged 18–32 years completed the questionnaire. Students were invited to participate if they were undertaking nursing, pharmacy, public health, and laboratory sciences and if they fulfilled the inclusion criteria: no previous pregnancy, of reproductive age, over the age of 18 years. The questionnaire was in English and Nepali.

The study received ethical approval at the Institutional Review Committee of Nepal Health Co-operative Ltd. (NEHCO-IRC) at Manmohan Memorial Institute of Health Sciences (MMIHS) in Kathmandu, Nepal. NEHCO-IRC/078/589. MMIHS IRC is accredited by NHRC, the governing body for ethical approval. The Co- PI of the research is from MMIHS, and the study site is MMIHS. The MMIHS IRC committee has gone through the standardised process and provided ethical approval.

A paper-based survey was administered (S1 Appendix). Participants who volunteered to complete the survey provided consent by completing the survey. All data collected and analysed was anonymised. Recruitment was from 15/08/2022 to 15/11/2022.

## Survey items

The questionnaire comprised the pain catastrophising scale (PCS) [16] and previous pain experiences, which have previously been used in Nepal [17]. The PCS consists of 13 items, rated on a 5-point Likert scale ranging from 0–4 with a total score ranging from 0 (no catastrophising) to 52 (severe pain catastrophising). The scale has high test-retest correlation (r = 0.75) across six weeks and good internal consistency of the three subscales with the total PCS (Cronbach's alpha = 0.87) [16]. The three subscales are: helplessness (questions 1–5,12), magnification (questions 6,7,13) and rumination (questions 8–11). The subscale helplessness relates to an inability to cope with pain. Magnification links with an overemphasised or amplified response to pain. While rumination refers to the negative thoughts that a person focuses on when they think of their past and present pain that evoke emotional distress. Permission was sort from the Mapi Research Trust for the use of the PCS. The Mapi Research Trust is a non-profit organisation dedicated to improving patient outcomes [27].

In addition, participants were asked to provide information around age, ethnicity, and religion. They were asked contextual questions about previous pain and current pain and pain sites. Questions about their period pains included whether they needed to take medication for their period pains and if their period pains affected their daily activities. The aim of the paper was to explore the prevalence of pain catastrophising and identify predictors for pain catastrophising rather than attitudes to menstruation. A Visual Analogue Scale (VAS) 0–10 was provided to report period pain intensity.

## Statistical analysis

Data were collated and organised in Microsoft Excel and analysed using Statistical Package for Social Sciences (SPSS v 28). Descriptive statistics were produced, and the data were checked for normality using histograms, Q-Q plots and the statistical tests of the PCS score ranges and pain intensity score. There was some evidence to suggest that the PCS score and pain intensity scores were not normal based on the significance, histogram (with chosen bin size) and the shape of the Q-Q plot. This data is reported using medians, and interquartile ranges. Age was reported in age ranges. Ethnicity and Religion were categorised. Prevalence of pain catastrophising and analysis were categorised and carried out using two cut-off scores the first at a PCS cut off $\geq$ 20 and above [24,28] and at the PCS cut off $\geq$ 30 and above [16,24]. The Pearson's chi square ($\chi2$) test was used to test the difference in distribution between the categorical variables. We conducted Binary logistic regression to examine the associations between the independent variables age and ethnicity with cut off scores of $\geq$ 20 and $\geq$ 30 and the two dependent variables (1) painful periods; (2) painful periods affecting daily activity. Overall, 98% of the records were complete, missing data ranged from 2.4%-0% across all variables.

## Results

### Descriptive analysis

A total number of 170 healthy nulliparous women who were completing undergraduate degree or MSc programs completed the survey (Table 1). Most women were aged 18–22 years (89.9%) and were Hindu (95.3%). While over 80% reported their ethnicity as either Brahmin, Chhetri or Newar. Participants' PCS scores were not associated with age, ethnicity, or religion.

A summary of descriptive data relating to the prevalence of pain catastrophising is presented (Table 2) at two cut off scores PCS $\geq$20 and PCS $\geq$30. The mean PCS for all participants is reported alongside the mean scores for the three subgroups and the mean period pain intensity score.

**Table 1. Demographic and descriptive information for study participants (N = 170).**

|  | Category | N (%) | PCS ≥20 n = 75 P (χ 2-sided) | PCS ≥30 n = 29 P (χ 2-sided) |
|---|---|---|---|---|
| Age (N = 169) | 18–22 years | 152 (89.9%) |  |  |
|  | 23–27 years | 16 (9.5%) |  |  |
|  | 28–32 years | 1 (0.6%) | 0.449 | 0.076 |
| Ethnicity (N = 170) | Chhetri | 38 (22.4%) |  |  |
|  | Brahmin | 65 (38.2%) |  |  |
|  | Newar | 35 (20.6%) |  |  |
|  | Other | 32 (18.8%) | 0.684 | 0.959 |
| Religion(N = 170) | Hindu | 162 (95.3%) |  |  |
|  | Buddhist | 7 (4.1%) |  |  |
|  | Other | 1 (0.6%) | 0.528 | 0.833 |

PCS Pain Catastrophising Scale.

The prevalence of pain catastrophising reported at a cut off score of PCS ≥20 was 55.9% (95/170) and at a cut off score of PCS ≥30 was 17.1% (29/170). The median and interquartile ranges (IQR) of participants total PCS scores were 9 (IQR 10–27). The median pain intensity scores were 7 (IQR 5–8). Pain symptoms associated with a PCS ≥20 and PCS<20 reported (Table 3).

Most participants reported painful periods 89%, and they were significantly more likely to have a PCS ≥20 (p <0.05) (Table 3). The duration of pain during a period was reported as lasting from one day to the length of the period with the majority (45%) reporting pain lasting two days. The percentage of participants who took pain relief was 35%, and there was a non-significant trend for those with a PCS ≥20 (p = >0.07) not to take pain relief (Table 3).

Almost two thirds of participants reported that period pain affected their lives (65%). Participants with a PCS≥ 20 were significantly more likely to report that period pains affected their daily life (p<0.001) (Table 3).

A total of 11% reported previous pain experiences and 24% reported current pain experiences. The most common site of both previous pain and current pain was the low back. Neither of these variables were associated with a PCS ≥20. Pain symptoms associated with a PCS ≥30 and PCS<30 are reported (Table 4).

**Table 2. Pain Catastrophising Scores (PCS), prevalence of pain catastrophising and mean total PCS scores, the three subscales (Rumination, Magnification, Helplessness) period pain intensity score N = 170.**

| PCS recorded at two cut-offs | Number and percentage at each cut-off score |
|---|---|
| PCS ≥20 | 95 (55.9%) |
| PCS ≥30 | 29 (17.1%) |
|  | **Median (IQR)** |
| **Total PCS (score range 0–52)** | 19.00 (10–27) |
| **PCS Subscale scores** |  |
| Rumination (score range 0–16) | 8 (3–11) |
| Magnification (score range 0–12) | 4 (2–6) |
| Helplessness (score range 0–24) | 6 (3–10) |
| **Period Pain intensity (range 0–10)** | 7(5–8) |

PCS Pain Catastrophising Scale; IQR Interquartile range.

**Table 3. Comparison of two groups PCS ≥20 and PCS<20 and reporting of period pain symptoms, current and previous pain.** (N = 170).

| N (%) | PCS ≥20 (%) | PCS <20 (%) | P χ 2-sided |
|---|---|---|---|
| **Painful periods**<br>Yes n = (89%) | 72 (96%) | 80 (84%) | |
| No n = (11%) | 3 (4%) | 15 (16%) | <0.05§ |
| **Period Pain medication taken**<br>Yes n = (35%) | 32 (32%) | 28 (39%) | |
| No n = (65%) | 67 (68%) | 43 (61%) | 0.074 |
| **Period Pain affecting daily life**<br>Yes n = (65%) | 60 (81%) | 50 (53%) | |
| No n = (35%) | 14 (19%) | 45 (47%) | <0.001 |
| **Previous pain experiences**<br>Yes n = (11%) | 11 (15%) | 8 (9%) | |
| No n = (89%) | 63 (85%) | 86 (91%) | 0.197 |
| **Current pain experiences**<br>Yes n = (24%) | 19 (26%) | 21 (23%) | |
| No n = (76%) | 54 (74%) | 72 (77%) | 0.606 |

§ Fishers Exact 2-sided; PCS Pain Catastrophising Scale.

All women with a PCS ≥30 reported having painful periods and this association was statistically significant (p<0.05) (Table 4). Participants who reported that period pain did not affect their daily lives were significantly more likely to report a PCS <30 (p<0.05). There was no association between those who reported period pain medication taken and a PCS ≥30 (p = 0.11), or current pain and PCS ≥30 (p = 0.28). There was a non-significant trend (p = 0.06) showing that those with a PCS ≥30 were more likely to have had previous pain.

## Interpretative analysis

Participants with a PCS ≥20 were 5.2 times [95%CI 1.40 to 19.70] more likely to report painful periods than those with a PCS of <20 (p<0.01). There were no associations between age and

**Table 4. Comparison of two groups PCS ≥30 and PCS<30 and their reporting of period pain symptoms and current and previous pain.** (N = 170).

| | PCS ≥30 | PCS <30 | P χ 2-sided |
|---|---|---|---|
| **Painful periods**<br>Yes n = (89%) | 29 (100%) | 123 (87%) | |
| No n = (11%) | 0 (0%) | 18 (13%) | <0.05 § |
| **Period pain medication taken**<br>Yes n = (35%) | 14 (48%) | 46 (33%) | |
| No n = (65%) | 15 (52%) | 95 (67%) | 0.108 |
| **Period Pain affecting daily life**<br>Yes n = (65%) | 23 (82%) | 87 (62%) | |
| No n = (35%) | 5 (18%) | 54 (38%) | <0.05 |
| **Previous pain**<br>Yes n = (11.3%) | 6 (21%) | 13 (9%) | |
| No n = (88.7%) | 22 (79%) | 127 (91%) | 0.064 |
| **Current pain**<br>Yes n = (24%) | 9 (32%) | 31 (22%) | |
| No n = (76%) | 19 (68%) | 107 (78%) | 0.275 |

§ Fishers Exact 2-sided; PCS Pain Catastrophising Scale.

painful periods. Participants of Brahmin origin were 4.5 times [95% CI 1.20 to 16.88] (p<0.05) and those of Newar origin were 5.7 times [1.07 to 30.53] (p<0.05) more likely to report painful periods. All women with a PCS ≥30 reported having painful periods.

Tables 5 and 6 report factors associated with painful periods affecting the daily lives of women reporting a PCS ≥20 and PCS ≥30 respectively.

Participants with a PCS≥20 were 4.0 times [95%CI 1.93 to 8.42] more likely to report that painful periods affected their daily activities this was statistically significant p <0.001 (Table 5). Ethnicity and age were not significantly associated.

A similar pattern was seen in relation to a PCS ≥30 and painful periods (Table 6). In this model participants with a PCS ≥30 were 3.4 times more likely [95%CI 1.10 to 10.53] to report that period pains affected their daily activities (p<0.03). Ethnicity and age were not significantly associated.

## Discussion

This is the first study to report the prevalence of pain catastrophising in a group of nulliparous women from Nepal. The prevalence of pain catastrophising at a cut off PCS ≥20 was higher in this study than in a methodologically similar study carried out amongst nulliparous UK university students (56% compared to 47%) [24]. In contrast nulliparous women in Nepal from this study had a slightly lower prevalence at a cut off PCS ≥30 compared to nulliparous women in the UK study (17% compared to 21%) [24]. This has implications for women and their babies as high pain catastrophising has been reported as a contributing factor to a rise in elective caesarean rates [22]. For women to receive useful support during pregnancy there is a requirement for health professionals to understand that those who have high pain catastrophising are also likely to have a fear of pain and pain related anxiety [24] and consider how these traits may be mitigated in order to improve birth outcomes.

This current study also explored factors that were associated with pain catastrophising. For those who reported PCS ≥20 and PCS ≥30 they were significantly more likely to have reported having painful periods, and in the PCS ≥20 group this factor was much more likely in those of Brahmin and Newar origin. However, when PCS ≥20 and PCS ≥30 were compared to those who reported the broader remit of 'painful periods which affected their daily lives', this factor was not associated with ethnicity. There are no direct comparisons in Nepal. Although a recent cohort study (n = 520) reported higher median PCS scores in Nepalese women compared with

**Table 5. Binary regression exploring predictors of painful periods that affect daily life with three independent variables; PCS ≥20, ethnicity, and age N = 168.**

|  | B | S.E | Wald | df | p | Exp(B) | 95% Confidence interval |
|---|---|---|---|---|---|---|---|
| PCS≥20 | 1.395 | 0.375 | 13.804 | 1 | <0.001* | 4.034 | 1.933 to 8.419 |
| Ethnicity |  |  | 3.103 | 3 | 0.376 |  |  |
| Ethnicity 1 | -0.115 | 0.482 | 0.057 | 1 | 0.812 | 0.892 | 0.347 to 2.293 |
| Ethnicity2 | -0.668 | 0.535 | 1.656 | 1 | 0.198 | 0.503 | 0.176 to 1.433 |
| Ethnicity 3 | -0.682 | 0.541 | 1.588 | 1 | 0.208 | 0.506 | 0.175 to 1.460 |
| Age |  |  | 0.802 | 2 | 0.670 |  |  |
| Age 1 | -0.521 | 0.582 | 0.802 | 1 | 0.370 | 0.594 | 0.190 to 1.858 |
| Age 2 | -23.086 | 40192.970 | 0.000 | 1 | 1.000 | 0.000 | 0.000 |
| Constant | 0.489 | 0.410 | 1.418 | 1 | 0.234 | 1.630 |  |

Variables entered on Step 1: PCS≥20 (Pain Catastrophising Score) Baseline; **Ethnicity** baseline Chhetri, (1) Brahmin; (2) Newar; (3) Other. **Age** baseline 18–22 years; (1) 23–27 years; (2) 28–32 years. $R^2$ = 0.13 (Cox & Snell) 0.17 (Nagelkerke), $\chi2$ (6) = 22.514, p< 0.001

* <0.05 Significant.

**Table 6. Binary regression exploring predictors of painful periods that affect daily life with three independent variables; PCS ≥30, ethnicity, and age N = 168.**

|            | B       | S.E       | Wald  | df | p      | Exp(B) | 95% Confidence interval |
|------------|---------|-----------|-------|----|--------|--------|-------------------------|
| PCS≥30     | 1.224   | 0.577     | 4.501 | 1  | 0.034* | 3.399  | 1.098 to 10.528         |
| Ethnicity  |         |           | 4.074 | 3  | 0.254  |        |                         |
| Ethnicity 1| -0.131  | 0.466     | 0.079 | 1  | 0.778  | 0.877  | 0.352 to 2.187          |
| Ethnicity2 | -0.759  | 0.516     | 2.161 | 1  | 0.142  | 0.468  | 0.170 to 1.288          |
| Ethnicity 3| -0.768  | 0.526     | 2.132 | 1  | 0.144  | 0.464  | 0.166 to 1.301          |
| Age        |         |           | 0.470 | 2  | 0.791  |        |                         |
| Age 1      | -0.386  | 0.563     | 0.470 | 1  | 0.493  | 0.680  | 0.226 to 2.049          |
| Age 2      | -23.306 | 40192.970 | 0.000 | 1  | 1.000  | 0.000  | 0.000                   |
| Constant   | 0.880   | 0.382     | 5.292 | 1  | 0.021  | 2.411  |                         |

Variables entered on Step 1: PCS≥30 (Pain Catastrophising Score) Baseline; **Ethnicity** baseline Chhetri, (1) Brahmin; (2) Newar; (3) Other. **Age** baseline 18–22 years; (1) 23–27 years; (2) 28–32 years. $R^2$ = 0.073 (Cox & Snell) 0.100 (Nagelkerke), $\chi2$ (6) = 12.645, p< 0.05;

\* <0.05 Significant.

a cohort in the United States, and it observed that those of Brahmin origin to be at greatest risk of developing chronic pain [29] but the authors explained that their study did not have the scope to explain this finding. There is little evidence from this study to suggest that ethnicity and pain catastrophising are linked but this may be an area for future research that explores a larger more diverse population.

Worry is a central component of pain catastrophising and it is suggested that the link with chronic pain stems from the continued vigilance to the threat of the pain [30]. It is well acknowledged that many women suffer with period pains through life. It is possible that in some cultures the pain experienced on a regular cycle is re-enforced and augmented by associations with cultural myths and taboos [18,31]. These myths and taboos may have important influences on health and well-being as they are embedded in health beliefs, behaviours, and perceptions. Which may impact on the way in which pain is communicated. Therefore, health professionals need to be open to listening to pain stories of some, while recognising that others may need more encouragement in sharing their story.

Women's experiences of pain in childbirth have been reported to be associated with what they understand to be the meaning of the pain. For example, if a woman interprets her pain to be related to something productive and meaningful, she may also report being better able to manage her pain [32]. While those who associate their pain with negative thoughts are more likely to seek external support to help them manage their pain. However, it was interesting in our current study that women who had higher PCS scores were less likely to report taking pain relief medication compared with those who had lower PCS scores. This maybe because pain medication is helpful for those who have mild period pain rather than those that report more severe pain. A Cochrane review found that although non-steroidal anti-inflammatory drugs (NSAIDs) are more effective than placebo for alleviating pain, nearly half of women across the studies felt they did not achieve adequate pain relief for their dysmenorrhea [33]. There are several factors that may modulate women's pain experiences and their fear of pain. These may because previously they have experienced delays in receiving pain medication [34] and they may perceive that the pain they report might not be taken seriously [35].

Women's experiences of pain may be one factor that is influencing the change in caesarean rates that have increased threefold from 2006–2016 in Nepal [36]. Among the different ethnic groups, the Brahmin/Chhetri had a higher caesarean rate increasing from 4.4% in 2006 to 11.3% in 2016. While those of Newar/Janajati origin have increased caesarean rates from 2.3%

in 2006 to 10.2% in 2016 [36]. Several authors have suggested that pain catastrophising and fear of childbirth are associated with rising rates of caesarean section [21,22]. Nulliparous women that experience fear in childbirth are three times more likely to have caesarean sections and that rate increases to fourfold for multiparous women. Although it is acknowledged that fear in childbirth may manifest during pregnancy [37], there is evidence that pain catastrophising manifests much earlier in a woman's life [24] and might contribute to negative thoughts that self-perpetuate fear and anxiety around pain through life.

It might be suggested that women's experiences early in life of not receiving adequate pain relief [32], coupled with the fact that pain may not be taken seriously [35] may be contributing to their negative emotions and beliefs about pain. It is suggested that there should be a commitment to train health professionals in Nepal about the importance of asking women about their previous pain experiences during their antenatal appointments. This would provide time to raise women's awareness about pain and strategies they can practice enabling them to successfully self-manage their pain in latent labour.

## Study strengths and limitations

The strengths of this study are that it is the first study to explore the prevalence of pain catastrophising in nulliparous women in Nepal and to explore the association of pain catastrophising with period pains. The self-reporting nature of the study means that there may be some recall bias, although period pains are a regular occurrence and therefore this recall is frequent. The participants were from a convenience sample of university students of which 90% were between the ages of 18–22 years studying on health-related programs, and so this limits the generalisability of the findings to other populations and further research is required. Most of the participants reported coming from a few ethnicities and the survey was of an urban population again reducing the generalisability to other populations. It is recognised that there are negative attitudes towards menstruation in Nepal, but this is unlikely to have affected the responses that were anonymous. It was not the aim of the paper to address attitudes to menstruation, instead it was to understand about pain.

## Conclusions

The study found a high prevalence of pain catastrophising in young nulliparous women from one higher education institution in Katmandu, Nepal. Women with higher pain catastrophising scores were more likely to report painful periods and to report that period pains affected their daily activities. Notably, women with higher pain catastrophising scores were less likely to take pain medication, which may be because they find pain medication to be ineffective. As women's previous negative experiences of pain relief may contribute to their fears of childbirth, it could be argued that pain catastrophising is contributing to the rise in caesarean births in Nepal. We suggest that health professionals should be educated about the multiple factors that contribute to a woman's perceptions and fears about pain through life, including their experiences of pain relief.

## Supporting information

**S1 Appendix. Survey questions.**
(PDF)

## Acknowledgments

We would like to acknowledge the contribution of Ansha KC for her support in data collection and data entry.

## Author Contributions

**Conceptualization:** Carol J. Clark, Vanora A. Hundley.

**Data curation:** Carol J. Clark, Vanora A. Hundley.

**Formal analysis:** Carol J. Clark, Sujan Babu Marahatta, Vanora A. Hundley.

**Funding acquisition:** Carol J. Clark.

**Methodology:** Carol J. Clark, Vanora A. Hundley.

**Project administration:** Sujan Babu Marahatta.

**Supervision:** Sujan Babu Marahatta.

**Writing – original draft:** Carol J. Clark.

**Writing – review & editing:** Sujan Babu Marahatta, Vanora A. Hundley.

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
