## [Decision Letter · Decision Letter 0]

14 Feb 2024

PONE-D-23-39925The prevalence of pain catastrophising in nulliparous women in Nepal; the importance for childbirthPLOS ONE

Dear Dr. Clark,

Thank you for submitting your manuscript to PLOS ONE. After careful consideration, we feel that it has merit but does not fully meet PLOS ONE’s publication criteria as it currently stands. Therefore, we invite you to submit a revised version of the manuscript that addresses the points raised during the review process. Please submit your revised manuscript by Mar 30 2024 11:59PM. If you will need more time than this to complete your revisions, please reply to this message or contact the journal office at plosone@plos.org. Please include the following items when submitting your revised manuscript:A rebuttal letter that responds to each point raised by the academic editor and reviewer(s). You should upload this letter as a separate file labeled 'Response to Reviewers'.A marked-up copy of your manuscript that highlights changes made to the original version. You should upload this as a separate file labeled 'Revised Manuscript with Track Changes'.An unmarked version of your revised paper without tracked changes. You should upload this as a separate file labeled 'Manuscript'.

We look forward to receiving your revised manuscript.

Kind regards,

Umesh Raj Aryal, PhD

Academic Editor

PLOS ONE

Journal Requirements:

3. In the online submission form, you indicated that "Complete de-identified data set is available for research purposes on application to the corresponding author."

**Additional Editor Comments:**

In this study, the Author needs to explain the operational definition of Nulliparous students.

It is not easy to identify any nulliparous student who experienced miscarriages, abortion, or stillbirth. The author needs to explain "how they access this information from students.

Data were collected online and on paper both and also in English and Nepali. In this situation reliability and validity will be questionable how will you ensure it?

"high test-retest correlation (r=0.75) across six weeks". How many times survey were performed for this correlation? This information should be explained in the methodology.

Table 1 presents the chi-square test which is unclear about a demographic profile with PCS >=30 0r 20

There is only 1 other religion. can you define other?

Boarders in Tables 3 and 4 make it difficult to identify p values. It is unclear whether "χ 2-sided". Present exact p value at P<0.05 unless p is very small.

In table 5, age 2 has zero values. discuss it.

Explain the significance of logistic regression in this study. many variables are not associated with PCS score.

Some results in the table and text do not match. Needs to recheck and the odds ratio with a wide interval range is questionable. It is because of the sample size.

In overall methodology and analysis is not clear needs to improve it. Consult statistician.

Reviewers' comments:

Reviewer's Responses to Questions

**Comments to the Author**

1. Is the manuscript technically sound, and do the data support the conclusions?

Reviewer #1: Yes

Reviewer #2: Yes

Reviewer #3: Yes

2. Has the statistical analysis been performed appropriately and rigorously? 

Reviewer #1: Yes

Reviewer #2: Yes

Reviewer #3: Yes

3. Have the authors made all data underlying the findings in their manuscript fully available?

Reviewer #1: Yes

Reviewer #2: Yes

Reviewer #3: Yes

4. Is the manuscript presented in an intelligible fashion and written in standard English?

Reviewer #1: Yes

Reviewer #2: Yes

Reviewer #3: Yes

5. Review Comments to the Author

Reviewer #1: 1. It is better to put the line number in the manuscript to make easier for the reviewer.

2. Authors should make more clear and readable foe the following sentence in the Introduction:

In Nepal pain

catastrophising has been recognised as a characteristic that influences post-operative pain intensity

in addition it was reported that women were found to have significantly higher PCS scores than men

and a higher PCS was correlated with increases in requirements for pain relieving medication [25].

3. Authors should clarify following regarding Ethical Approval in Study Design and Participants section:

- How the Institutional Review committee at MMIHS could provide ethical approval to the study? Authors need to explain the responsibiity of the committee to clear the ethical approval for this study.

4. In Discussion section:

- The study additionally revealed that there is no significant relationship between a high PCS score (>30) and medication usage. The authors need to provide further elaboration on how these findings should be interpreted, particularly in connection with PCS scores exceeding 20 and their association with medication utilization.

Reviewer #2: Thank you very much for providing an opportunity for reviewing the article entitled “The prevalence of pain catastrophising in nulliparous women in Nepal; the importance for childbirth”. The authors are grateful for providing the information about prevalence of pain catastrophizing in nulliparous women in Nepal. It is really interesting and new for Nepal. But there are some comments which are provided directly in the text as sticky note. The article has no any line numbers, hence difficult to indicate the comments in separate pages. The authors are suggested to improve their methods and be clear in the results and discussion sections. The articles may be accepted if the authors addressed all the comments provided in the MS directly. I recommend the major revision of the article.

Reviewer #3: 1. The topic selected is good but sample is too low to generalize it in context of Nepal. Only a college is selected in capital city of Nepal which also seems to be related to health science that often possesses a good knowledge and skill. As a prevalence study, it doesn’t indicate real status of women in colleges in local and peripheral especially rural area women. Neither can it present real status of Kathmandu only as the college selected is itself a private where usually students from a rich family background study pursue higher education. Authors are suggested to review topic or mention properly about this methodology portion.

2. Please give the rationale of selecting urban based private college which is itself a health science college. Why did you ignored the status of women of non-health science, public colleges and rural areas who usually are from back warded socio economic level.

3. It is better to place keywords in abstract in alphabetical order.

4. The study design selected seems to be cross sectional but the author has mentioned prospective. Please explain in detail about design adopted.

5. In a reputed journal like Plos one, use of proper statistical language for population and sample needs to be distinguished. The author has ignored such minor aspects.

6. Authors are advised to keep the meaning of other attributes in demographic and other information.

6. PLOS authors have the option to publish the peer review history of their article (what does this mean?). If published, this will include your full peer review and any attached files.

Reviewer #1: **Yes: **Basant Adhikari

Reviewer #2: No

Reviewer #3: No

---

## [Author Response · Author response to Decision Letter 0]

9 May 2024

Please find attached Response to Reviewers - which provides a table of responses.

Questions relating to reviewers 2 and 3 are amalgamated under Reviewer 2. this includes the 'sticky notes'.

---

## [Editor Report · Decision Letter 1]

23 May 2024

PONE-D-23-39925R1The prevalence of pain catastrophising in nulliparous women in Nepal; the importance for childbirthPLOS ONE

Dear Dr. Clark,

Thank you for submitting your revised manuscript to PLOS ONE. After careful consideration, we feel that it has merit but does not fully meet PLOS ONE’s publication criteria as it currently stands. Therefore, we invite you to submit a revised version of the manuscript that addresses the points raised during the review process.

**ACADEMIC EDITOR: **

Authors need to provide an operation definition of "nulliparous women"? Still an unclear reason for choosing nulliparous women in the introduction section.

How did you conform to normality? Just mentioning a histogram is not enough. Either present histogram or value of skewness.

Did the Author check the reliability of all three subscales?

In the Introduction, the objective of the study:

to explore the prevalence of pain catastrophizing in nulliparous women in Nepal and to identify predictors for pain catastrophizing.

In the Method Section, the aim of the study:

 to explore perceptions of pain rather than attitudes to menstruation.

These two statements look different. can you please clarify it?

The Pearson’s chi-square (χ2) test was used to test the difference in distribution between the categorical variables. What is the significance of using this study?

Participants’ PCS scores were not associated with age, ethnicity, or religion. Table is one way and explaining association. Please justify it.

There is no heading for Yes/No Response in Table 2. 

 6.39 [SD 2.39] (0-10)- remove SD 

Exp(B)- Different to understand to the reader who does not have a statistical background. Some values are zero. Please consult a statistician for logistic regression.  

95% Confidence interval- 1.933 to 8.419- replace to by -.The performance for the Logistic regression is unclear and just taking age and ethnicity. 

Consult a Statistician for data analysis.

In conclusion

"The study found a high prevalence of pain catastrophizing in young nulliparous women in Nepal". How studying one college represents Nepal. Next results are repeated in the conclusion. so the conclusion needs to be revised. 

We look forward to receiving your revised manuscript.

Kind regards,

Umesh Raj Aryal, PhD

Academic Editor

PLOS ONE

Additional Editor Comments:

Authors need to provide an operation definition of "nulliparous women"? Still an unclear reason for choosing nulliparous women in the introduction section.

How did you conform to normality? Just mentioning a histogram is not enough. Either present histogram or value of skewness.

Did the Author check the reliability of all three subscales?

In the Introduction, the objective of the study:

to explore the prevalence of pain catastrophizing in nulliparous women in Nepal and to identify predictors for pain catastrophizing.

In the Method Section, the aim of the study:

to explore perceptions of pain rather than attitudes to menstruation.

These two statements look different. can you please clarify it?

The Pearson’s chi-square (χ2) test was used to test the difference in distribution between the categorical variables. What is the significance of using this study?

Participants’ PCS scores were not associated with age, ethnicity, or religion. Table is one way and explaining association. Please justify it.

There is no heading for Yes/No Response in Table 2.

6.39 [SD 2.39] (0-10)- remove SD

Exp(B)- Different to understand to the reader who does not have a statistical background. Some values are zero. Please consult a statistician for logistic regression.

95% Confidence interval- 1.933 to 8.419- replace to by -.The performance for the Logistic regression is unclear and just taking age and ethnicity.

Consult a Statistician for data analysis.

In conclusion

"The study found a high prevalence of pain catastrophizing in young nulliparous women in Nepal". How studying one college represents Nepal. Next results are repeated in the conclusion. so the conclusion needs to be revised.

---

## [Author Response · Author response to Decision Letter 1]

4 Jun 2024

Reviewers comments Authors Response

Reviewer 3 Thank you for the helpful and constructive comments. Please find highlighted changes in the revised manuscript.

Authors need to provide an operation definition of "nulliparous women"? Still an unclear reason for choosing nulliparous women in the introduction section. Definition of nulliparous p 4

This part of the introduction has be re-written: 

The pattern of labour pain differs …….P5 

and their pain management requirements.

How did you conform to normality? Just mentioning a histogram is not enough. Either present histogram or value of skewness. The advice of a statistician was sought and amendments to the text p7 using histograms, Q-Q plots…. and Table 2 on p9

Did the Author check the reliability of all three subscales? The reliability of the three subscales were not explored in this paper as this was not the purpose. The subscales have been reported descriptively for context and are not used in any interpretative tests. Table 2 P9

In the Introduction, the objective of the study:

to explore the prevalence of pain catastrophizing in nulliparous women in Nepal and to identify predictors for pain catastrophizing.

In the Method Section, the aim of the study:

 to explore perceptions of pain rather than attitudes to menstruation.

These two statements look different. can you please clarify it?

 This is clarified.

The objective of the study remains as stated.

This was re-written in the methods section in response to a previous reviewer below in this reviewer/author response document p7* and has been further clarified in the text of the paper on p7

The aim of the paper was to explore the prevalence of pain catastrophising and identify predictors for pain catastrophising rather than attitudes to menstruation.

The Pearson’s chi-square (χ2) test was used to test the difference in distribution between the categorical variables. What is the significance of using this study?

 Statistical support was sought.

Pearson’s chi square test is being used to test if there is an association between the categorical variables. In table three for example exploring differenced between those with PCS ≥20 and PCS< 20 and reporting pain perceptions/symptoms.

Participants’ PCS scores were not associated with age, ethnicity, or religion. Table is one way and explaining association. Please justify it.

 For clarity we have included the table and the associated text to support different readers needs.

There is no heading for Yes/No Response in Table 2. 

 6.39 [SD 2.39] (0-10)- remove SD 

 The heading for Table 2 has been altered to simplify on P9. 

SD removed.

Exp(B)- Different to understand to the reader who does not have a statistical background. Some values are zero. Please consult a statistician for logistic regression. 

The performance for the Logistic regression is unclear and just taking age and ethnicity. 

Consult a Statistician for data analysis.

 A statistician was consulted for this analysis.

95% Confidence interval- 1.933 to 8.419- replace to by -. The 95% Confidence intervals have been reported with a ‘to’ instead of – to avoid confusion with any negative numbers.

In conclusion

"The study found a high prevalence of pain catastrophizing in young nulliparous women in Nepal". How studying one college represents Nepal. Next results are repeated in the conclusion. so the conclusion needs to be revised. 

 For clarity this has been re-written p 17

from one higher education institution in Kathmandu, Nepal.

The next section has been re-written p17Women with higher pain catastrophising scores……

---

## [Decision Letter · Decision Letter 2]

18 Jul 2024

The prevalence of pain catastrophising in nulliparous women in Nepal; the importance for childbirth

PONE-D-23-39925R2

Dear Dr. Clark,

We’re pleased to inform you that your manuscript has been judged scientifically suitable for publication and will be formally accepted for publication once it meets all outstanding technical requirements.

Kind regards,

Rabie Adel El Arab

Academic Editor

PLOS ONE

Additional Editor Comments (optional):

Reviewers' comments:

Reviewer's Responses to Questions

**Comments to the Author**

1. If the authors have adequately addressed your comments raised in a previous round of review and you feel that this manuscript is now acceptable for publication, you may indicate that here to bypass the “Comments to the Author” section, enter your conflict of interest statement in the “Confidential to Editor” section, and submit your "Accept" recommendation.

Reviewer #1: All comments have been addressed

Reviewer #2: All comments have been addressed

2. Is the manuscript technically sound, and do the data support the conclusions?

Reviewer #1: Yes

Reviewer #2: Yes

3. Has the statistical analysis been performed appropriately and rigorously? 

Reviewer #1: Yes

Reviewer #2: Yes

4. Have the authors made all data underlying the findings in their manuscript fully available?

Reviewer #1: Yes

Reviewer #2: Yes

5. Is the manuscript presented in an intelligible fashion and written in standard English?

Reviewer #1: Yes

Reviewer #2: Yes

6. Review Comments to the Author

Reviewer #1: (No Response)

Reviewer #2: Dear authors

Thank you authors for your hard work. All the comments addressed or answered technically. Authors should menimized the typo errors in this MS. I hope this articles will be baseline for the further research. Hence, I recommended for the publication.

7. PLOS authors have the option to publish the peer review history of their article (what does this mean?). If published, this will include your full peer review and any attached files.

Reviewer #1: **Yes: **Basant Adhiari

Reviewer #2: **Yes: **Dr. Jagan Nath Adhikari

---

## [Editor Report · Acceptance letter]

26 Jul 2024

PONE-D-23-39925R2 

PLOS ONE

Dear Dr. Clark, 

I'm pleased to inform you that your manuscript has been deemed suitable for publication in PLOS ONE. Congratulations! Your manuscript is now being handed over to our production team.

Kind regards, 

on behalf of

Dr. Rabie Adel El Arab 

Academic Editor

PLOS ONE